# Refined Mixed-Strategy Multi-modal Representation Learning for Recommendation

## Abstract

Representations of users and items are crucial in recommendation systems as they capture the latent relation of user-item interactions. Recent multi-modal recommendation models leveraging multi-modal features have effectively improved recommendation performance. However, existing methods lack the exploration of multiple potential relations between users and multi-modal items effectively. This paper introduces an approach called Refined Mixed-Strategy Multi-modal recommender (RMSM), which aims to model the interaction between users and multi-modal items by creating refined multi-modal heterogeneous graphs and learning multiple relations through mix-strategy. Specifically, we construct an interaction graph that encompasses diverse modalities between users and items. Subsequently, RMSM leverages mix-strategy across multiple graph types to derive comprehensive representations. Extensive experiments on three public datasets show that RMSM can achieve the best results compared with baselines. Numerous ablation studies and visualization are performed on RMSM to confirm its efficacy in the context of recommendation, specifically focusing on its performance within heterogeneous and homogeneous sub-graphs.

## 1 Introduction

Recommendation systems have become a necessary tool for users to find products and services of their choice. Recommendation is still a significant and challenging task that has received substantial attention from the academic and industry community. For instance, recommendation methods based on Collaborative Filtering (CF) techniques, such as Matrix Factorization (MF) Koren et al. (2009); Zhang et al. (2023b), have been developed to learn efficiency representations for users and items. They aim to predict future interactions based on historical interactions. Although many studies have been developed to develop reliable and efficient methods for recommendations, these existing studies still need to solve the cold-start issue Zhou et al. (2023c). That is, the scenario needs to deal with new users and items. At the same time, their representations have yet to be learned due to the limited interactions.

Most previous works rely on multi-modal information from users and items to alleviate the cold-start issue in recommendation. Recent multi-modal recommendation systems (MMRec) based on multi-modal information to improve the recommendation performance have gained considerable attention Zhang et al. (2023a). Some works use multi-modal features as side information to improve the items' representation based on traditional CF methods. With the development of graph-based recommendation systems, recent works focus on modeling user-item interactions with Graph Neural Networks (GNNs). For instance, MMGCN Wei et al. (2019) conducts personalized micro-video recommendation via user-item graph, the items' representations derived from item-item graph of each modality. Graph-based recommendation performance heavily relies on the quality of user-item interaction graph. While the interacted items of the specific user always consist uninterested items, such interactions between users and uninterested items are named false-positive interactions. To refine the structure of the interaction graph, GRCN Wei et al. (2020b) is proposed to discover potential false-positive interactions. LATTICE Zhang et al. (2021b) mines the latent item-item collaborative item-item relations through high-order item-user-item relations and achieves considerable performance. Based on LATTICE, MICRO Zhang et al. (2022) uses the contrastive learning schema to model the modality-shared and modality-specific representation space and achieve better recommendation performance. FREEDOM Zhou & Shen (2023) freezes and denoises the interaction

graph simultane for multi-modal recommendation. DRAGON Zhou et al. (2023a) enhances the interactive relations by learning dual representations of both users and items via constructing homogeneous graphs. While current multi-modal recommendation methods aggregate the multi-modal representations of items through pre-fusion and enhance item representations by leveraging their multi-modal properties, they effectively mitigate the challenges posed by cold-start users and items. However, modeling the heterogeneous relations between users and multi-modal items, as well as the sparse interactions between cold-start users and items, remains essential.

Heterogeneous graphs containing diverse node and connection types are ubiquitous in real-world applications. It is an important method to alleviate the sparsity of cold start interaction. Representation learning on such graphs aims to encode node embeddings that capture semantic heterogeneity and relational richness Yang et al. (2022). To enable this, HGCL Chen et al. (2023) incorporates heterogeneous semantics into user-item modeling via contrastive learning across views. The enriched heterogeneous semantics enhance user and item representations, mitigating cold-start sparsity. Similarly, IHGNN Cai et al. (2023) harnesses heterogeneous relations in cold-start recommenders to enrich sparse user attributes. These studies demonstrate the utility of modeling heterogeneous, relational information to alleviate data sparsity. Outstanding challenges remain in effectively capturing hidden, complex semantics and further enriching user representations.

In fact, modeling heterogeneous relations using heterogeneous graphs mostly relies on meta-path methods. For instance, HeRec [9] extracts node information based on different meta-paths in heterogeneous graphs and combines this information with MF to improve the performance of personalized recommendations. The performance of meta-path based recommendation models depends on the quality of manually designed meta-paths, and it is not possible to model the different impacts of different types of nodes on the current node when aggregating the content of adjacent nodes. Therefore, HetGNN [10] learns heterogeneous node representations by integrating heterogeneous node type information.

Besides, the existing MMRec methods mostly construct latent item-item graph of each modality and fuse them to the final item-item graph. It is apparent that such methods only focus on the interactions between user representation and final multi-modal item representation. While the interactions between users and different item representations contain rich and informative heterogeneous relations. Such multi-modal heterogeneous relations also need to be considered. Therefore, in modeling users and multi-modal information using Heterogeneous Graph Neural Networks (HGNNs), how to consider the heterogeneity of nodes and the relation between multi-modal attributes to generate high-quality representations is also a problem that needs to be considered. The challenges of MMRec can be conclude as follows:

- Pre-fusing multi-modal representations of an item might inadvertently overshadow the distinctive traits inherent in each modality, potentially resulting in a diminished sense of modality independence within the item representations.

- Besides, how to measure the heterogeneity and homogeneity of nodes and the different impacts of multi-modal attributes, as well as the relations between multi-modal attributes, to generate high-quality representations is the second challenge.

Towards this end, we propose a framework named RMSM that learns multiple relations of different modalities for multi-modal recommendation. Specifically, we construct the heterogeneous interaction graph between users and different items with different modalities. Existing building Multimodal recommenders mostly represent items with different modalities and aggregate multi-modal representations to extract item interaction features. RMSM constructs the interactions between users and items with different modalities. The relations between users and different modal item representations are represented in a fine-grained manner. Then, we design the relation learning module with mix-strategy to learn multiple relations, including heterogeneous and homogeneous relation learning process. Finally, we aggregate the multiple relation representations by GCN to conduct recommendations. We conduct experiments on three experiments to prove the superiority of RMSM. Experimental results show that RMSM achieves SOTA performance on both benchmarks. Extensive experiments further show that the heterogeneous and homogeneous relations will contribute to the overall improvement and make the learned representations more effective. Our contributions are as follows:

- We highlight the significance of modeling heterogeneous and homogeneous relations through a refined multi-modal graph structure, and build a refined graph that includes multiple latent adjacency relations.

- We propose a multi-modal recommendation framework named RMSM, which adopts a mixed-strategy approach to model and learn the explicit and implicit relations between users and multi-modal items.

- RMSM achieves state-of-the-art performance on three benchmarks. We also provide ablation studies and visualizations for embedding space to prove its effectiveness.

## 2 RELATED WORK

### 2.1 MULTI-MODAL RECOMMENDER

Collaborative Filtering (CF) has achieved great success in recommendation systems, which leverage users' feedbacks (such as clicks and purchases) to predict the preferenceof users and make recommendations. However, CF-based methods suffer from sparse data with limited user-item interactions and rarely accessed items. To address the problem of data sparsity, it is important to exploit other information besides user-item interactions. Multi-modal recommendation systems consider massive multi-media content information of items, which have been successfully applied to many applications, such as e-commerce, instant video platforms and social media platforms. For example, VBPR He & McAuley (2016) enhances matrix factorization by integrating visual attributes derived from product images, aiming to elevate the overall performance of the recommendation system. DVBPR Kang et al. (2017) attempts to jointly train the image representation as well as the parameters in recommender. DeepStyle Liu et al. (2017) separates category information from visual representations to effectively learn the stylistic characteristics of items and capture user preferences. VECF Chen et al. (2019) captures the diverse attentions of users towards distinct image regions and reviews through its modeling approach.

Recently, Graph Neural Networks (GNNs) have been introduced into recommenders and especially multi-modal recommenders. MMGCN Wei et al. (2019) establishes dedicated graphs for each modality and applies graph convolutional operations to grasp modality-specific user preferences and concurrently refine item representations. In this way, the learned user representation can reflect the users' specific interests on items. Continuing the path set by MMGCN, GRCN (Graph Refinement and Convolution Network) Wei et al. (2020a) places emphasis on dynamically enhancing the structure of the interaction graph to uncover and eliminate possible false-positive edges. There are several prior studies Wei et al. (2022); Liu et al. (2023) that propose to explore collaborative item relations through high-order item-user-item co-occurrences. For example, HUIGN (Hierarchical User Intent Graph Network) Wei et al. (2022) constructs a co-interacted item graph that showcases users' intents across various levels, with the objective of deriving multi-level user intents from item co-interaction patterns and subsequently improving recommendation effectiveness. MGCL Liu et al. (2023) captures collaborative signals stemming from interactions and leverages visual and textual modalities to individually extract user preference cues specific to each modality.

### 2.2 HETEROGENEOUS GRAPH LEARNING

In real-world applications, heterogeneous graphs are prevalent, featuring diverse node types and connections. The objective of representation learning on heterogeneous graphs is to encode node embeddings that effectively retain the intricate semantics encompassing relation diversity, as emphasized by Yang et al. Yang et al. (2020). In pursuit of this objective, HGNNs emerge as promising methodologies that offer cutting-edge representation outcomes. For example, HAN Wang et al. (2019b) enhances the graph attention network with the capability of dealing with heterogeneous types of nodes and relations based on meta-path construction. Taking inspiration from the transformer architecture Vaswani et al. (2017), HGT (Heterogeneous Graph Transformer) Hu et al. (2020) devises a graph transformer network to facilitate heterogeneous message propagation. It achieves this by employing self-attention to compute propagation weights between nodes. Moreover, MAGNN Fu et al. (2020) takes into account both intra- and inter-metapath aggregation to merge information originating from distinct meta-paths within heterogeneous graphs. In the context of HGIB (Heterogeneous Graph Information Bottleneck) Yang et al. (2021), the concept of

information bottleneck is expanded to encompass heterogeneous graph learning, incorporating self-supervision mechanisms within homogeneous graphs.

Many existing methods for constructing multi-modal recommenders based on heterogeneous graphs primarily rely on multi-modal information fusion to merge representations of items across different modalities, subsequently modeling interaction relations between users and items. However, these methods often overlook the significance of user representations in bridging the modality gap between different modal representations of items and neglect the heterogeneous relations between user and item modal representations. This paper aims to address this gap by focusing on modeling the heterogeneous relations between user and item modal representations, thereby establishing a connection between the modality gap of various item representations through user representations.

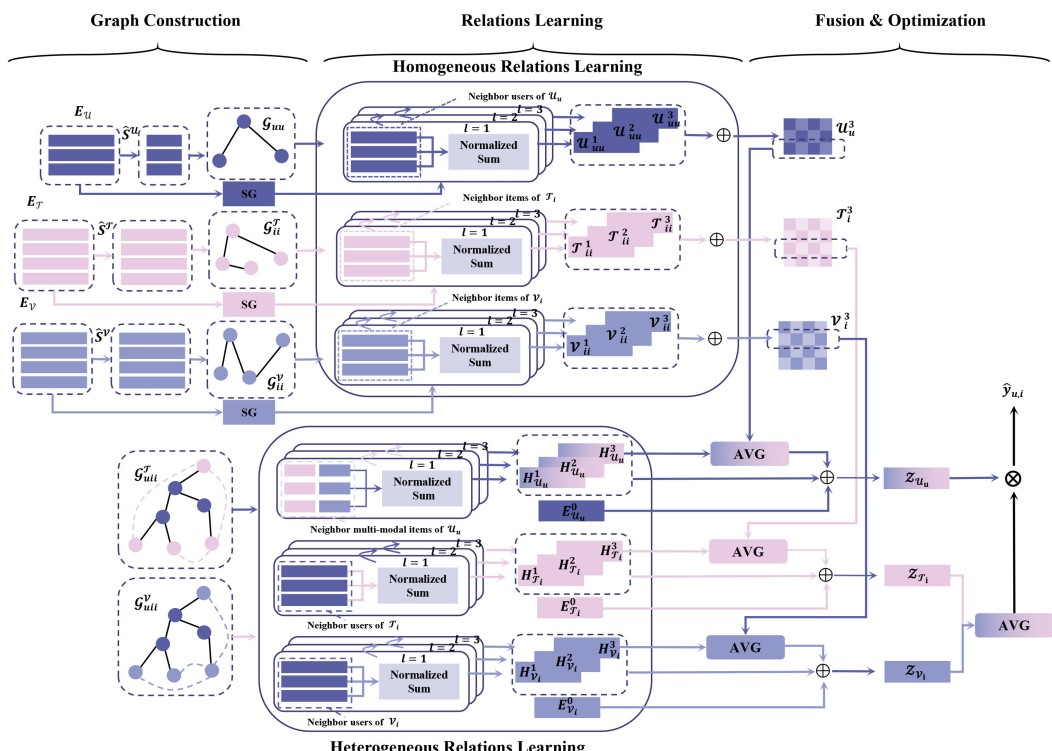

Figure 1: The framework of RMSM. RMSM includes three key components, multi-modal heterogeneous graph construction, heterogeneous relation learning, feature fusion and optimization.

## 3 PRELIMINARIES

Here, we describe the definition of Multi-Modal Recommendation (MMRec) task. Given the set of users $\mathcal{U}$, the item set $\mathcal{I} = \{\mathcal{T}, \mathcal{V}\}$ with distinct modalities, where $\mathcal{T}$ and $\mathcal{V}$ denote the textual and visual modality item sets, respectively. $\mathcal{G}_{ui} = \{\mathcal{U}, \mathcal{I}, \varepsilon_{ui}\}$ signifies the interaction graph between user $u \in \mathcal{U}$ and item $i \in \mathcal{I}$. It contains $\mathcal{G}_{ui}^{\mathcal{T}}$ and $\mathcal{G}_{ui}^{\mathcal{V}}$ subgraphs, representing the interactive connections between users and items of different modalities. For these subgraphs, we characterize adjacent matrices $\mathbf{A}_{ui}^{\mathcal{T}}, \mathbf{A}_{ui}^{\mathcal{V}} \in \mathbb{R}^{M \times N}$, matching graph $\mathcal{G}_{ui}^{\mathcal{T}}$ and $\mathcal{G}_{ui}^{\mathcal{V}}$, in turn. Here, $M$ and $N$ represent the count of users and items, respectively. The goal of MMRec is to forecast unobserved interactions among users and multi-modal items, given the graphs with relation heterogeneity. The goal of MMRec is to predict the likelihood of interaction between users and multi-modal items. $\mathcal{G}_{uu} = \{\mathcal{U}, \varepsilon_{uu}\}$ is defined to represent user relations among users, it inclues user-wise social connections with the edge set $\varepsilon_{uu}$. To incorporate different modality item-wise relations, we define the item interaction graph $\mathcal{G}_{ii}^{\mathcal{T}} = \{\mathcal{T}, \varepsilon_{ii}^{\mathcal{T}}\}$ and $\mathcal{G}_{ii} = \{\mathcal{V}, \varepsilon_{ii}^{\mathcal{V}}\}$ to connect dependent items with different modalties, respectively. For these defined graphs, we define four adjacent matrices $\mathbf{A}_{ui}^{\mathcal{T}}, \mathbf{A}_{ui}^{\mathcal{V}} \in \mathbb{R}^{M \times N}$, $\mathbf{A}_{uu} \in \mathbb{R}^{M \times M}$, $\mathbf{A}_{ii}^{\mathcal{T}} \in \mathbb{R}^{N \times N}$ and $\mathbf{A}_{ii}^{\mathcal{V}} \in \mathbb{R}^{N \times N}$, corresponding to graph $\mathcal{G}_{ui}^{\mathcal{T}}, \mathcal{G}_{ui}^{\mathcal{V}}, \mathcal{G}_{uu}, \mathcal{G}_{ii}^{\mathcal{T}}$ and $\mathcal{G}_{ii}^{\mathcal{V}}$, respectively.

# 4 METHODOLOGY

In this section, we elaborate on our proposed RMSM, as shown in Figure 1. RMSM mainly consists of three modules: Multi-modal heteroheneous graph initialization, modality-aware heterogeneous graph learning, and traing optimization.

## 4.1 MULTI-MODAL HETEROHENEOUS GRAPH INITIALIZATION

Prior to capturing the collaborative signal within the multi-modal heterogeneous graph, it is imperative to forge the multi-modal heterogeneous graph through the utilization of interaction data among users and multi-modal items, along with the multi-modal attributes of the items. The construction process encompasses both the user and item feature initialization procedure and the multi-modal heterogeneous graph construction process.

### 4.1.1 EMBEDDING INITIALIZATION

For the embedding of users, denoted as $\mathcal{U} = \{u_1, u_2, ..., u_M\}$ are employed, skillfully initialized through the application of the xavier initializer. The dimension size of user is $d$. Concerning the item embeddings, which belong to diverse modalities, represented as $\mathcal{T} = \{t_1, t_2, ..., t_N\}$ and $\mathcal{V} = \{v_1, v_2, ..., v_N\}$. The dimension size of textual and visual items is $d^{\mathcal{T}}$ and $d^{\mathcal{V}}$, respectively. Subsequently, for the sake of ease and coherence, the features sourced from various modalities are mapped into a shared representation space of equal dimensionality, that is the dimension size of $t_i$ and $v_i$ is $d$. The formulation of the initial embedding matrices for both users is $\mathbf{E}_{\mathcal{U}} \in \mathbb{R}^{M \times d}$, $\mathbf{E}_{\mathcal{T}} \in \mathbb{R}^{N \times d}$, and $\mathbf{E}_{\mathcal{V}} \in \mathbb{R}^{N \times d}$.

### 4.1.2 MULTI-MODAL HETEROHENEOUS GRAPH CONSTRUCTION

To consistently and effectively consider users, items, their respective attributes, and different associated relation information, we build Multi-modal Heteroheneous Graph (MHG), denoted as $\mathcal{G}_{\text{MHG}} = (\mathcal{C}, \mathcal{R})$, where $\mathcal{C} = \mathcal{U} \cup \mathcal{T} \cup \mathcal{V}$ and $\mathcal{R} = \mathcal{R}_{\text{uu}} \cup \mathcal{R}_{\text{tt}} \cup \mathcal{R}_{\text{vv}} \cup \mathcal{R}_{\text{ut}} \cup \mathcal{R}_{\text{uv}}$ represent the node set and relation set of $\mathcal{G}_{\text{MHG}}$. $\mathcal{R}_{\text{uu}}$ represents the relations between users. $\mathcal{R}_{\text{tt}}$ represents the relations between textual characteristics of items. $\mathcal{R}_{\text{vv}}$ represents the relations between visual characteristics of items. $\mathcal{R}_{\text{ut}}$ represents the relations between users and textual characteristics of items. $\mathcal{R}_{\text{uv}}$ represents the relations between users and visual characteristics of items. In MMRec, the interaction data provide the relations $\mathcal{R}_{\text{ut}}$ and $\mathcal{R}_{\text{uv}}$. For realtions $\mathcal{R}_{\text{uu}}$, $\mathcal{R}_{\text{tt}}$ and $\mathcal{R}_{\text{vv}}$, we design following methods to exploit them, and we introduce it with $\mathcal{R}_{\text{uu}}$ as example.

Based on the assumption of homogeneity theory, it is known that users/items with similar characteristics are more likely to interact. We quantify the characteristic relation between two users/items by their similarity. The similarity matrix $S^m \in \mathbb{R}^{N \times N}$ is compted by

$$S_{ij}^{\mathcal{U}} = \frac{(u_i)^{\top}(u_j)}{||u_i||||u_j||}. \tag{1}$$

Then we conduct $k$-NN sparsification on the dense graph: for each user, we only keep edges with the top-$k$ confidence scores:

$$\hat{S}_{ij}^{\mathcal{U}} = \begin{cases} S_{ij}^{\mathcal{U}}, & S_{ij}^{\mathcal{U}} \in \text{top-}k(s_i^{\mathcal{U}}), \\ 0, & \text{otherwise}, \end{cases} \tag{2}$$

here, $\hat{S}^{\mathcal{U}}$ represents the outcome of the sparsified. To mitigate the issue of gradient explosion or vanishing, we normalize the adjacency matrix as follows:

$$\tilde{S}^{\mathcal{U}} = (D^{\mathcal{U}})^{-\frac{1}{2}} \hat{S}^{\mathcal{U}} (D^{\mathcal{U}})^{-\frac{1}{2}}, \tag{3}$$

where $D^{\mathcal{U}} \in \mathbb{R}^{N \times N}$ is the diagonal degree matrix of $\hat{S}^{\mathcal{U}}$ and $D_{ii}^{\mathcal{U}} = \sum_j \hat{S}_{ij}^{\mathcal{U}}$. Here, we get the $\mathcal{R}_{\text{uu}}$, and then, we use the above method to construct $\mathcal{R}_{\text{tt}}$ and $\mathcal{R}_{\text{vv}}$.

## 4.2 MODALITY-AWARE HETEROGENEOUS GRAPH LEARNING

Here, we get the modality-aware heterogeneous graph. To effectively model rich, heterogeneous, and hidden relations in modality-aware heterogeneous graphs, we conduct graph learning from two views, including homogeneous relation learning and heterogeneous relation learning.

### 4.2.1 HOMOGENEOUS RELATIONS LEARNING

There exist three types of homogeneous relations, user-user $\mathcal{R}_{uu}$, textual item-item $\mathcal{R}_{tt}$ and visual item-item $\mathcal{R}_{vv}$. To get the different homogeneous relation representations, the representations of user and multi-modal item are fed into different graph encoders. For graph encoder, we employ LightGCN He et al. (2020) and the message propagation. To accentuate the disparities in interaction patterns among different relation types, we utilize the self-gating module Yu et al. (2021) to generate modality-aware embeddings for user-level social connections and item-level semantic relations. These embeddings are derived from the shared initial embedding space and are represented as follows:

$$\mathcal{U}^0 = \mathbf{E}_{\mathcal{U}}^0 \odot \sigma\left(\mathbf{E}_{\mathcal{U}}^0 \mathbf{W}_{\mathcal{U}} + \mathbf{b}_{\mathcal{U}}\right); \mathcal{T}^0 = \mathbf{E}_{\mathcal{T}}^0 \odot \sigma\left(\mathbf{E}_{\mathcal{T}}^0 \mathbf{W}_{\mathcal{T}} + \mathbf{b}_{\mathcal{T}}\right); \mathcal{V}^0 = \mathbf{E}_{\mathcal{V}}^0 \odot \sigma\left(\mathbf{E}_{\mathcal{V}}^0 \mathbf{W}_{\mathcal{V}} + \mathbf{b}_{\mathcal{V}}\right), \tag{4}$$

where $\mathcal{U}^0 \in \mathbb{R}^{M \times d}$, $\mathcal{T}^0 \in \mathbb{R}^{N \times d}$, and $\mathcal{V}^0 \in \mathbb{R}^{N \times d}$ are the embeddings for the homogeneous graphs $\mathcal{G}_{uu}$, $\mathcal{G}_{ii}^{\mathcal{T}}$ and $\mathcal{G}_{ii}^{\mathcal{V}}$ for different heterogeneous relations, respectively. $\sigma(\cdot)$ denotes the sigmoid activation function. $\odot$ denotes element-wise multiplication operation. $\mathbf{W}_m \in \mathbb{R}^{d \times d}$ and $\mathbf{b}_m \in \mathbb{R}^{d \times 1}$ are the transformation and bias parameters, $m \in \{\mathcal{U}, \mathcal{T}, \mathcal{V}\}$. Through the self-gating mechanism with multiplicative skip-connection, embeddings $\mathcal{U}^0$, $\mathcal{T}^0$, and $\mathcal{V}^0$ not only share common semantics with initial embeddings $\mathbf{E}_{\mathcal{U}}^0$, $\mathbf{E}_{\mathcal{T}}^0$, and $\mathbf{E}_{\mathcal{V}}^0$, but also acquire the versatility to delineate both user-user interactions and multi-modal item-item relations.

Among the above embedding matrices are used as input for the user-user view and the multi-modal item-item view, respectively. To obtain isomorphic relations hidden in $\mathcal{G}_{uu}$, $\mathcal{G}_{uu}^{\mathcal{T}}$ and $\mathcal{G}_{uu}^{\mathcal{V}}$, we employ LightGCN He et al. (2020) as the encoder for the these views of graph structures. For U-U, textual I-I and visual I-I views, the message propagation of $\mathcal{G}_{uu}$, $\mathcal{G}_{ii}^{\mathcal{T}}$ and $\mathcal{G}_{ii}^{\mathcal{V}}$ can be formulated as:

$$\mathcal{U}^{l+1} = \sum_{u \in \mathcal{N}_u^{\mathcal{U}}} \frac{1}{|\mathcal{N}_u^{\mathcal{U}}|}\left(\mathcal{U}^l\right); \mathcal{T}^{l+1} = \sum_{i \in \mathcal{N}_i^{\mathcal{T}}} \frac{1}{|\mathcal{N}_i^{\mathcal{T}}|}\left(\mathcal{T}^l\right); \mathcal{V}^{l+1} = \sum_{i \in \mathcal{N}_i^{\mathcal{V}}} \frac{1}{|\mathcal{N}_i^{\mathcal{V}}|}\left(\mathcal{V}^l\right), \tag{5}$$

where $\mathcal{N}_u^{\mathcal{U}}$ means the neighbors users of user $u$. $\mathcal{N}_i^{\mathcal{T}}$ and $\mathcal{N}_i^{\mathcal{V}}$ mean the textual and visual neighbors of item $i$, respectively.

### 4.2.2 HETEROGENEOUS RELATIONS LEARNING

When users select products, the multi-modal properties of the product can affect their decision-making, and the contribution of different modal product content to the final decision is also different. In order to capture the heterogeneous relation between different modal representations of users and items, we utilize user-item interaction data to supplement the relation between users and different modal contents of the item. It is worth noting that we use different graph encoders for user-item interaction diagrams with different modalities to obtain representations of user-item content in different modalities. For modality-aware user-item interaction graphs $\mathcal{G}_{ui}^{\mathcal{T}}$ and $\mathcal{G}_{ui}^{\mathcal{V}}$, the message propagation of heterogeneous message ropagation can be formulated as:

$$\mathbf{H}_{\mathcal{U}_u}^{l+1} = \sum_{i \in \mathcal{N}_u^{\mathcal{U}}} \sum_{u \in \mathcal{N}_i^{\mathcal{T}, \mathcal{V}}} \frac{1}{\sqrt{|\mathcal{N}_u^{\mathcal{U}}|}\sqrt{\left|\mathcal{N}_i^{\mathcal{T}, \mathcal{V}}\right|}}\left(\mathbf{H}_{\mathcal{T}_i}^l + \mathbf{H}_{\mathcal{V}_i}^l\right)\mathbf{H}_{\mathcal{U}_u}^l,$$

$$\mathbf{H}_{\mathcal{T}_i}^{l+1} = \sum_{u \in \mathcal{N}_i^{\mathcal{T}}} \sum_{i \in \mathcal{N}_u^{\mathcal{U}}} \frac{1}{\sqrt{\left|\mathcal{N}_i^{\mathcal{T}}\right|}\sqrt{|\mathcal{N}_u^{\mathcal{U}}|}}\mathbf{H}_{\mathcal{U}_u}^l \mathbf{H}_{\mathcal{T}_i}^l, \tag{6}$$

$$\mathbf{H}_{\mathcal{V}_i}^{l+1} = \sum_{u \in \mathcal{N}_i^{\mathcal{V}}} \sum_{i \in \mathcal{N}_u^{\mathcal{U}}} \frac{1}{\sqrt{\left|\mathcal{N}_i^{\mathcal{V}}\right|}\sqrt{|\mathcal{N}_u^{\mathcal{U}}|}}\mathbf{H}_{\mathcal{U}_u}^l \mathbf{H}_{\mathcal{V}_i}^l,$$

where $\mathcal{N}_u$ and $\mathcal{N}_i$ denote the neighbor set of target nodes $u$ and $i$, respectively. $\mathbf{H}_{\mathcal{U}_u}^l$, $\mathbf{H}_{\mathcal{T}_i}^l$, and $\mathbf{H}_{\mathcal{V}_i}^l$ denote the representation of user $u$, textual item $i$ and visual item $i$ in $l$-th iteration, respectively. $\mathbf{H}_{\mathcal{U}_u}^0 = \mathbf{E}_{\mathcal{U}_u}$, $\mathbf{H}_{\mathcal{T}_i}^0 = \mathbf{E}_{\mathcal{T}_i}$ and $\mathbf{H}_{\mathcal{V}_i}^0 = \mathbf{E}_{\mathcal{V}_i}$.

### 4.2.3 INFORMATION AGGREGATION

Taking cues from the soft meta-path concept introduced in Chen et al. (2023), the process involves aggregating information in each iteration from diverse heterogeneous relations. These high-order

embeddings effectively retain heterogeneous semantics by encompassing multi-hop connections across multiple rounds of heterogeneous message propagation. Notably, the embeddings for both users and items are iteratively updated following the prescribed heterogeneous fusion protocol:

$$\hat{\mathcal{U}}_u^{l+1} = f\left(\mathbf{H}_{\mathcal{U}_u}^{l+1}, \mathcal{U}_u^{l+1}\right); \hat{\mathcal{T}}_i^{l+1} = f\left(\mathbf{H}_{\mathcal{T}_i}^{l+1}, \mathcal{T}_i^{l+1}\right); \hat{\mathcal{V}}_i^{l+1} = f\left(\mathbf{H}_{\mathcal{V}_i}^{l+1}, \mathcal{V}_i^{l+1}\right), \tag{7}$$

where $\hat{\mathcal{U}}_u^{l+1} \in \mathbb{R}^{M \times d}$, $\hat{\mathcal{T}}_i^{l+1}$, $\hat{\mathcal{V}}_i^{l+1} \in \mathbb{R}^{N \times d}$ integrate heterogeneous semantics for the next layer. $f(\cdot)$ represents the fusion operation for heterogeneous information. In addition, for enhanced aggregation of heterogeneous information alongside encoded layer-specific representations ($1 \le l \le L$), we formulate the overall embeddings of users and items in the ensuing manner:

$$\mathcal{Z}_{\mathcal{U}_u} = \mathbf{E}_{\mathcal{U}_u}^0 + \sum_{l=1}^{L} \frac{\mathbf{H}_{\mathcal{U}_u}^l}{||\mathbf{H}_{\mathcal{U}_u}^l||}; \mathcal{Z}_{\mathcal{T}_i} = \mathbf{E}_{\mathcal{T}_i}^0 + \sum_{l=1}^{L} \frac{\mathbf{H}_{\mathcal{T}_i}^l}{||\mathbf{H}_{\mathcal{T}_i}^l||}; \mathcal{Z}_{\mathcal{V}_i} = \mathbf{E}_{\mathcal{V}_i}^0 + \sum_{l=1}^{L} \frac{\mathbf{H}_{\mathcal{V}_i}^l}{||\mathbf{H}_{\mathcal{V}_i}^l||}, \tag{8}$$

where $L$ denotes the maximum number of GCN iterations. We add the initial embeddings $\mathbf{E}_{\mathcal{U}_u}^0$, $\mathbf{E}_{\mathcal{T}_i}^0$, $\mathbf{E}_{\mathcal{V}_i}^0$ using skip connections. The expressions provided above depict the process of aggregating layer-specific representations for the user-item interaction perspective. Similarly, the embeddings for the U-U social perspective (denoted as $\mathcal{U}_{uu}$) and the I-I dependency perspective (i.e., $\mathcal{T}_{ii}$ and $\mathcal{V}_{ii}$) are acquired via analogous multi-order information aggregation procedures.

### 4.3 TRAING AND OPTIMIZATION

With the embeddings $\mathcal{U}_u$, $\mathcal{T}_i$ and $\mathcal{V}_i$, our method forecast the likelihood of user $u$ interacting with item $i$ via dot-product: $\hat{y}_{u,i} = \mathcal{Z}_{\mathcal{U}_u}^\top (\mathcal{Z}_{\mathcal{T}_i} + \mathcal{Z}_{\mathcal{V}_i})/2$, where $\mathcal{Z}_{\mathcal{U}_u}$, $\mathcal{Z}_{\mathcal{T}_i}$ and $\mathcal{Z}_{\mathcal{V}_i}$ denote the final embedding vectors of user $u$ and multi-modal item $i$. For each training sample, we maximize the prediction score as follows:

$$\mathcal{L}_{\text{BPR}} = \sum_{(u,i^+,i^-)} -\ln(\text{sigmoid}(\hat{y}_{u,i^+} - \hat{y}_{u,i^-})) + \lambda||\Theta||^2, \tag{9}$$

where $\lambda$ denotes a hyperparameter to determine the weight of the regularization term.

Table 1: overall performance comparison. Bold represents optimal value, underline represents sub-optimal value. Larger value means the better performance.

| Model | Baby | | | Clothing | | | Sports | | |
|---|---|---|---|---|---|---|---|---|---|
| | R@20 | P@20 | N@20 | R@20 | P@20 | N@20 | R@20 | P@20 | N@20 |
| NGCF Wang et al. (2019a) | 0.0591 | 0.0032 | 0.0261 | 0.0387 | 0.0020 | 0.0168 | 0.0728 | 0.0038 | 0.0332 |
| LightGCN He et al. (2020) | 0.0698 | 0.0037 | 0.0319 | 0.0470 | 0.0024 | 0.0215 | 0.0803 | 0.0042 | 0.0377 |
| SGL Wu et al. (2021) | 0.0440 | 0.0024 | 0.0200 | 0.0191 | 0.0010 | 0.0088 | 0.0430 | 0.0023 | 0.0202 |
| VBPR He & McAuley (2016) | 0.0946 | 0.0050 | 0.0409 | 0.0853 | 0.0044 | 0.0392 | 0.1061 | 0.0053 | 0.0483 |
| MMGCN Wei et al. (2019) | 0.0646 | 0.0036 | 0.0276 | 0.0352 | 0.0018 | 0.0148 | 0.0631 | 0.0036 | 0.0267 |
| GRCN Wei et al. (2020a) | 0.0825 | 0.0046 | 0.0363 | 0.0660 | 0.0034 | 0.0285 | 0.0915 | 0.0052 | 0.0406 |
| MVGAE Yi & Chen (2022) | 0.0091 | 0.0005 | 0.0027 | 0.0057 | 0.0003 | 0.0017 | 0.0033 | 0.0002 | 0.0016 |
| DualGNN Wang et al. (2023) | 0.0807 | 0.0045 | 0.0353 | 0.0696 | 0.0036 | 0.0305 | 0.0899 | 0.0050 | 0.0404 |
| LATTICE Zhang et al. (2021a) | 0.0829 | 0.0044 | 0.0368 | 0.0710 | 0.0036 | 0.0316 | 0.0915 | 0.0048 | 0.0424 |
| BM3 Zhou et al. (2023b) | 0.0883 | 0.0048 | 0.0383 | 0.0625 | 0.0033 | 0.0280 | 0.0980 | 0.0033 | 0.0438 |
| MICRO Zhang et al. (2022) | 0.0898 | 0.0047 | 0.0407 | 0.0824 | 0.0042 | 0.0371 | 0.1005 | 0.0052 | 0.0467 |
| FREEDOM Zhou & Shen (2023) | 0.0992 | 0.0054 | 0.0424 | 0.0941 | 0.0047 | 0.0420 | 0.1089 | 0.0056 | 0.0481 |
| DRAGON Zhou et al. (2023a) | 0.1021 | 0.0056 | 0.0435 | 0.0957 | 0.0050 | 0.0435 | 0.1124 | 0.0060 | 0.0500 |
| LGMRec Guo et al. (2024) | 0.1002 | - | 0.0440 | 0.0828 | - | 0.0371 | 0.1068 | - | 0.0480 |
| PGL Yu et al. (2025) | 0.1040 | - | 0.0448 | 0.1014 | - | 0.0451 | 0.1144 | - | 0.0509 |
| MIG-GT Hu et al. (2025) | 0.1021 | - | 0.0452 | 0.0934 | - | 0.0422 | 0.1130 | - | 0.0511 |
| RMSM | **0.1235** | **0.0062** | **0.0533** | **0.1389** | **0.0073** | **0.0682** | **0.1335** | **0.0075** | **0.0634** |

## 5 EXPERIMENTS

We conduct our experiments on three benchmarks, **Baby, Sports and Clothing** McAuley et al. (2015). These datasets both have visual and textual modalities to describe items. Three widely used

protocols are used to evaluate the performance of top-*n* recommendation. We compare the baseline models have been introduced in related works. Our experimental setup is structured to investigate the following research inquiries:

- RQ1: How does RMSM's performance compare to that of existing methods ?
- RQ2: Does the integration of key components within our RMSM lead to an improvement in recommendation performance ?
- RQ3: How does key hyperparameter affect model performance ?

## 5.1 OVERALL COMPARISON (RQ1)

To assess the superiority of RMSM, we conduct a comparative analysis with all baseline methods across the three datasets. The outcomes are presented in Table 1, revealing the subsequent observations:

- RMSM consistently exhibits notable performance enhancements when compared to the baseline methods. RMSM facilitates proficient knowledge transfer across heterogeneous relations, thus aiding in modeling user preferences effectively. The incorporation of multi-modal heterogeneous graphs, integrating various relations, significantly amplifies recommendation performance through diverse training signals.
- DRAGON, based on HGNNs, frequently demonstrates superior performance compared to alternative methods. This underscores the effectiveness of integrating heterogeneous relational knowledge and item semantic relevance from both user and item perspectives into recommendation systems.

Table 2: Results of ablation studies. Bold represents optimal value

| Model | Baby | | | Clothing | | | Sports | | |
|---|---|---|---|---|---|---|---|---|---|
| | R@20 | P@20 | N@20 | R@20 | P@20 | N@20 | R@20 | P@20 | N@20 |
| Only CF | 0.0627 | 0.0031 | 0.0215 | 0.0283 | 0.0012 | 0.0126 | 0.0786 | 0.0023 | 0.0243 |
| $\mathcal{G}_{\text{UI}}+\mathcal{G}_{\text{II}}$ | 0.0783 | 0.0042 | 0.0476 | 0.0702 | 0.0033 | 0.0301 | 0.0943 | 0.0048 | 0.0461 |
| $\mathcal{G}_{\text{UI}}+\mathcal{G}_{\text{UU}}+\mathcal{G}_{\text{II}}^{T}+\mathcal{G}_{\text{II}}^{V}$ | **0.1235** | **0.0062** | **0.0533** | **0.1389** | **0.0073** | **0.0682** | **0.1335** | **0.0075** | **0.0634** |

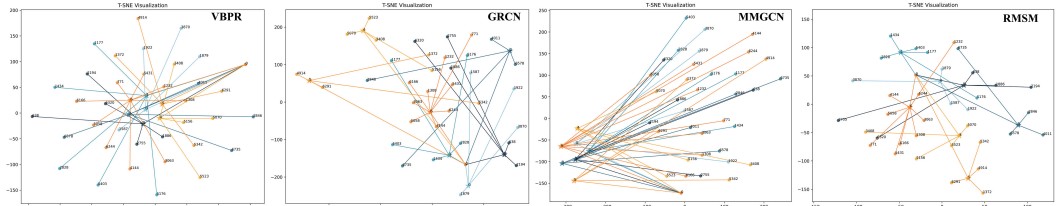

Figure 2: T-SNE visualization, stars mean the users and circles mean items. The left, middle and right part are VBPR, MMGCN, RMSM, respectively. As shown in the right part (RMSM), compared to other methods, the user representation obtained from RMSM training is more discriminative, and the representation of items that have interacted with the user is more distinct from the user representation.

## 5.2 ABLATION STUDIES OF MAIN MODULES (RQ2)

We use different CF methods based on different graphs to select the most suitable method for RMSM. The results are shown in the Table 2. We use LightGCN as final Collaborative Filtering (CF) component in RMSM. $\mathcal{G}_{\text{UI}}$ only use user item interaction data to model the user's preferences. $\mathcal{G}_{\text{UI}} + \mathcal{G}_{\text{II}}$ the latent interaction relation between items is used to model user preferences based on user item interaction data. $\mathcal{G}_{\text{UI}} + \mathcal{G}_{\text{UU}} + \mathcal{G}_{\text{II}}^{T} + \mathcal{G}_{\text{II}}^{V}$ use interaction relation between the different modal item representations, and they are put into the same heterogeneous, and the user's preferences are modeled using the multiple relations in the heterogeneous graph. The results shown in Table 2, we observe that:

- In $\mathcal{G}_{\mathrm{UI}} + \mathcal{G}_{\mathrm{II}}$, the homogeneous relation graph $\mathcal{G}_{\mathrm{II}}$ extracted from fused multi-modal information. Compared with $\mathcal{G}_{\mathrm{UI}}$, this manner get the better performance, and prove the effectiveness of multi-modal information and the latent homogeneous relations in recommendation. Multi-modal information provide external and rich information to model user preference.

- $\mathcal{G}_{\mathrm{UI}} + \mathcal{G}_{\mathrm{UU}} + \mathcal{G}_{\mathrm{II}}^{T} + \mathcal{G}_{\mathrm{II}}^{V}$ achieve best results prove the effectiveness of heterogeneous relations hidden in multi-modal heterogeneous interaction graph. This manner capture the fine-grained interaction features between users and items with different modality. It is also could be proved that the essential of focusing on different modality items.

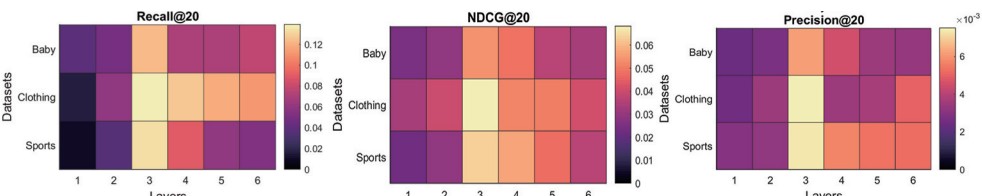

Figure 3: The bottom three images shows recommendation performance with different GCN layers.

## 5.3 HYPER-PARAMETER STUDIES (RQ3)

As illustrated in Figure 2, we delve into the impact of graph convolution layer depth. Investigating the influence of the number of graph convolution layers, we explore layers from $\{1, 2, 3, 4, 5, 6\}$. Notably, we observe a gradual enhancement in model performance as the number of graph convolution layers increases across the three datasets. This trend underscores the efficacy of capturing higher-order collaborative relations in more accurately modeling user preferences. However, for RMSM, performance begins to decline as the layer depth increases further. This phenomenon can be attributed to the potential over-smoothing issue associated with excessively deep graph convolution layers.

## 5.4 VISUALIZATION

In this section, we thoroughly examine the effectiveness of the representations produced by RMSM. To achieve this, we randomly selected seven users from the Baby dataset. Subsequently, we curated a collection of items that had not been paired during the training phase. Employing the t-SNE algorithm, we visualize these representations in a two-dimensional space. The visualizations of item and user representations, as derived from the VBPR, MMGCN, and RMSM models, are showcased in Figure 2.

Comparing these visualizations, we notice that in the RMSM model, stars and points of the same color, denoting users and their corresponding items, tend to be clustered more closely than in the MMGCN model. These distinctive clusters evident in the RMSM model's visualizations indicate its effectiveness in modeling user preferences.

## 6 CONCLUSION

This introduce RMSM to comprehend user and item representations by creating refined multi-modal interaction graphs and learning multiple relations based on mix-strategy. Specifically, we construct interaction graphs for user-user and item-item relations, encompassing distinct modalities. These interaction relations are amalgamated into a heterogeneous interaction graph. Subsequently, we establish a relation learning module to capture co-occurrence relations between users and a semantic graph for items that captures their semantic interrelations across various modalities. Through extensive experiments conducted on three distinct datasets, we effectively demonstrate the prowess of RMSM model.

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
