# OpenReview forum: "Refined Mixed-Strategy Multi-modal Representation Learning for Recommendation"
_ICLR.cc/2026/Conference — Submitted to ICLR 2026_

### Official Review · Reviewer_LdX7 · 2025-10-24

**Soundness:** 3
**Presentation:** 3
**Contribution:** 3
**Rating:** 6
**Confidence:** 3

**Summary:**

This paper proposes a multi-modal recommendation framework named RMSM, which integrates refined heterogeneous graph construction with mixed-strategy learning. The method enhances recommendation performance by explicitly modeling both homogeneous and heterogeneous relations within multi-modal data. The effectiveness is validated on multiple benchmark datasets, where it achieves state-of-the-art results.

**Strengths:**

(1) The paper introduces a multi-modal graph structure, designed to model heterogeneous and homogeneous relations by incorporating multiple latent adjacency relations.

(2) The work presents a mixed-strategy multi-modal recommendation framework aimed at modeling and learning both explicit and implicit user-item relations.

**Weaknesses:**

(1)  Please clarify the specific form of the aggregation function f(·) in Equation (7) and how it integrates the heterogeneous and homogeneous representations.

(2) To ensure reproducibility, it is recommended that the authors elaborate on the implementation details of the "different graph encoders" mentioned in Section 4.2.2. Clarifying the specific model architectures and any key hyperparameters used would be crucial.

**Questions:**

(1)The discussion of the experimental results in Section 5.1 should be expanded. Please provide a more causal analysis explaining the reasons behind the performance gains of RMSM over the baselines.

(2) To better assess the computational efficiency of the proposed method, please provide an analysis of its time complexity and/or a comparative study of the actual running time.

(3) It is recommended that the clarity of Figure 1 be enhanced. Specifically, the issues of overlapping elements and indistinct arrows should be addressed to ensure the figure effectively communicates the proposed framework.

(4) It is necessary to supplement the study with a sensitivity analysis of key hyperparameters, such as the value of $k$ in the k-NN graph sparsification and the regularization coefficient $\lambda$ in the BPR loss.

---

### Official Review · Reviewer_2AFj · 2025-10-27

**Soundness:** 3
**Presentation:** 3
**Contribution:** 2
**Rating:** 4
**Confidence:** 5

**Summary:**

This paper addresses limitations in existing multi-modal recommendation (MMRec) methods—insufficient user-multi-modal item relation exploration, modality independence loss from pre-fusion, and cold-start sparsity—by proposing RMSM (Refined Mixed-Strategy Multi-modal recommender). RMSM constructs a Multi-modal Heterogeneous Graph (MHG) with 5 relation types, learns homogeneous/heterogeneous relations via a mixed strategy (LightGCN, self-gating, dedicated encoders), and optimizes training.

**Strengths:**

1. RMSM constructs a multimodal heterogeneous graph that encompasses user–user, user–item across modalities (text/visual), and within‑modality item–item relations, enabling comprehensive modeling of user–item interactions while preserving modality‑specific signals that are often diluted by early fusion.

2. By decoupling the learning of homogeneous relations (user–user and within‑modality item–item, via LightGCN with a self‑gating module) from heterogeneous relations (cross‑modality user–item, via tailored message‑passing mechanisms), the method effectively integrates information from multiple sources.

**Weaknesses:**

1. Computational overhead and complexity: The framework integrates modules for multimodal heterogeneous graph construction (similarity computation, sparsification, normalization), dual‑path relation learning (homogeneous vs. heterogeneous), and multi‑round message aggregation. These stages involve substantial matrix operations and iterative message passing, which may increase training and inference costs. A formal complexity analysis and comparative runtime/memory experiments are needed.

2. Limited empirical scope and scalability: Experiments are restricted to three Amazon datasets (Baby, Sports, Clothing) with visual and textual modalities. The lack of evaluations on other domains (e.g., Netflix, TikTok) or larger‑scale benchmarks leaves generalization and scalability insufficiently validated.

3. Incomplete baselines and related‑work coverage: The paper omits comparisons or discussion of several closely related multimodal recommendation methods [1,2,3], making it difficult to contextualize the claimed improvements.

[1] Y. Wei, W. Liu, F. Liu, X. Wang, L. Nie, and T.-S. Chua. Lightgt: A light graph transformer for multimedia recommendation. In SIGIR, pages 1508–1517, 2023.

[2] Z.Tao, X.Liu, Y.Xia, X.Wang, L.Yang, X.Huang, and T.-S.Chua. Self-supervised learning for multimedia recommendation. Transactions on Multimedia (TMM), 2022.

[3] Jiang, Yangqin, Lianghao Xia, Wei Wei, Da Luo, Kangyi Lin, and Chao Huang. "Diffmm: Multi-modal diffusion model for recommendation." In Proceedings of the 32nd ACM International Conference on Multimedia, pp. 7591-7599. 2024.

**Questions:**

1. Can the authors provide some complexity analysis or comparative experiments of efficiency for RMSM?

---

### Official Review · Reviewer_hUvG · 2025-11-01

**Soundness:** 2
**Presentation:** 1
**Contribution:** 2
**Rating:** 2
**Confidence:** 4

**Summary:**

This paper introduces a graph-based framework designed to capture the complex relationships between users and the distinct modalities of items. Instead of fusing item modalities prematurely, RMSM constructs a comprehensive heterogeneous graph that includes not only user-item interactions but also homogeneous relations like user-user social connections and item-item semantic similarities for each modality. The model then employs a "mixed-strategy" learning approach, using different graph convolution operations to separately learn from homogeneous and heterogeneous relations, before aggregating these diverse signals to produce final user and item representations for recommendation.

**Strengths:**

- The "mixed-strategy" is a clever design choice. The model separates the learning of homogeneous relations (e.g., social influence via LightGCN) from heterogeneous relations (e.g., user-modality preference). This disentanglement allows the model to learn different types of collaborative and semantic signals more effectively, preventing them from being diluted in a single, one-size-fits-all message-passing scheme. The strong results of the ablation study (Table 2) clearly validate that modeling all these different graph structures is essential to its success.
- The paper demonstrates significant and consistent performance improvements over a wide range of strong baselines across three public datasets (Table 1). The inclusion of detailed ablation studies and t-SNE visualizations (Figure 2) provides robust evidence for the framework's effectiveness and helps to build confidence that the performance gains are directly attributable to its novel design.

**Weaknesses:**

- The paper lacks critical implementation details that are necessary for reproducing the results. Key hyper-parameters such as the learning rate, embedding dimensions, batch size, and the specific value of k used for the k-NN graph construction are not specified. Furthermore, without access to the source code, it is exceptionally difficult for other researchers to verify the findings or build upon the proposed framework.
- The experiments are conducted on three Amazon datasets that are relatively dense and of moderate scale, particularly after preprocessing. The true test for a multi-modal recommendation model is its ability to perform in large-scale, highly sparse environments where cold-start problems are most acute. Evaluating on widely-used, larger benchmarks such as MovieLens or the full Amazon Electronics dataset would be essential to more convincingly demonstrate the framework's effectiveness and scalability.
- The framework's primary drawback is its complexity, which raises concerns about scalability. It requires the construction and storage of multiple graphs (user-user, textual item-item, visual item-item, etc.). Building similarity-based graphs (like U-U) can be computationally prohibitive on large datasets, and the multi-view graph learning process is inherently more memory- and time-intensive than simpler models. This could make it challenging to deploy in production environments with millions of users and items.
- Many typos and grammatical errors, such as, Line 54: "graph simultane" etc.

**Questions:**

Please refer the weaknesses.

---

### Official Review · Reviewer_fH6C · 2025-11-03

**Soundness:** 3
**Presentation:** 2
**Contribution:** 1
**Rating:** 4
**Confidence:** 5

**Summary:**

This paper introduces RMSM (Refined Mixed-Strategy Multi-modal recommender) to address limitations in existing multi-modal recommendation systems. The authors argue that current methods use pre-fusion approaches that combine item modalities before modeling user interactions, losing modality-specific information. RMSM constructs heterogeneous graphs capturing separate relations between users and textual/visual item features, using a mixed-strategy to learn both homogeneous and heterogeneous relations. Experiments on three Amazon datasets show RMSM outperforms 16 baseline methods with comprehensive ablation studies confirming effectiveness.

**Strengths:**

1. The paper proposes a more fine-grained modeling method to capture the relations between users and items' multi-modal features, avoiding the information loss caused by pre-fusion approaches in existing methods.

2. The paper proposes a novel heterogeneous graph construction method for multi-modal recommendation that explicitly models separate relations between users and different item modalities rather than treating items as unified multi-modal entities.

3. The experimental evaluation is comprehensive, comparing against 16 diverse baseline methods including recent state-of-the-art approaches, and includes thorough ablation studies, hyperparameter analysis, and t-SNE visualizations that provide good insights into the model's effectiveness.

**Weaknesses:**

1. Though it is somehow new to use heterogeneous graph methods for multi-modal user behavior data, heterogeneous graph neural networks have been well-studied in previous works. The author should better clarify their technical novelty and contribution relative to existing works. The core technical components (LightGCN, self-gating mechanisms, similarity-based edge construction) are largely borrowed from existing literature, and the main contribution appears to be their specific combination rather than fundamental algorithmic innovations.

2. The paper argues that "existing methods lack the exploration of multiple potential relations between users and multi-modal items effectively" in the abstract. According to the paper's content, this refers to limitations including pre-fusion overshadowing modality-specific traits and insufficient modeling of heterogeneous user-modality relations. However, this statement itself is somewhat vague, and the specific problems seem like minor design choices rather than fundamental limitations. The authors should better refine this statement to provide clearer research motivation and more convincingly demonstrate why these are critical limitations rather than alternative design approaches.

**Questions:**

1. What specific technical innovations does your method contribute beyond applying existing HGNN techniques to multi-modal recommendation?

2. Can you provide concrete evidence demonstrating why pre-fusion and limited relation exploration are fundamental problems rather than reasonable design choices?

---

### Comment · Area_Chair_XgVg · 2025-11-22

Dear Reviewers,

Thank you for your time and effort in reviewing submissions for ICLR  2026. As we begin the author-reviewer discussion process, we kindly remind you to submit your responses to the author rebuttals by **December  2**.


Your engagement in this discussion phase is crucial to ensuring a fair and thorough evaluation of each submission.

**Action Required**


- Carefully consider the authors’ rebuttal and any additional evidence they provide.

- Update your review (if applicable) to reflect your revised perspective.

-  **Discuss with the authors if further details are required**


Your AC

---

### Meta-Review · Area_Chair_hTGG · 2025-12-03

**Summary:**

Based on the reviewers’ feedback and my own reading of the paper, the overall quality still needs improvement. We regret to inform you that this paper has not been accepted for this year’s conference. We hope the authors can address the relevant issues in subsequent revisions and achieve acceptance in future submissions.

**Reviewer Concerns:**

No rebuttal

**Reviewer Scores:**

The specific technical innovations do the method contributes beyond applying existing HGNN techniques to multi-modal recommendation should be provided.

---

### Decision · Program_Chairs · 2026-01-26

Reject